# An Update on Dynamic Changes in Cytokine Expression and Dysbiosis Due to Radiation Combined Injury

**DOI:** 10.3390/ijms262110456

**Published:** 2025-10-28

**Authors:** Juliann G. Kiang, Georgetta Cannon

**Affiliations:** 1Scientific Research Department, Armed Forces Radiobiology Research Institute, Uniformed Services University of the Health Sciences, Bethesda, MD 20814, USA; 2Department of Pharmacology and Molecular Therapeutics, Uniformed Services University of the Health Sciences, Bethesda, MD 20814, USA

**Keywords:** mouse model of radiation combined injury, cytokine, microbiome, survival, miR-34a, inflammation

## Abstract

The complexity of adverse responses from radiation injury (RI) followed by physical trauma, namely, radiation combined injury (RCI), is unique and more pronounced than either insult alone due to a poor understanding of the integration of these insults at the molecular/cellular/tissue and/or organ levels. It was shown that mice receiving ^60^Co γ-photon RCI with wounding had a lower LD_50/30_ than RI alone. This survival synergism was observed in bone marrow and the gastrointestinal system, as evidenced by an increase in γ-H2AX expression in bone marrow cell DNA, loss of circulatory blood cells, elevation of serum cytokine concentration, and activation of nuclear factor-κB/inducible nitric oxide synthase, and an earlier onset of bacterial infection and sepsis after RCI than after RI was detected. Dysbiosis (imbalance of the gut microbiota) was observed. There remains a pressing need for both prophylactic countermeasures and therapeutic remedies to deal with RCI threats. Investigations of how RCI can affect this important network of communication between the gut microbiota and other organs, including the brain, lung, heart, liver, kidney, and skin, could lead to new and critical interventions and prevention strategies. This review provides an update on new RCI animal models, dynamic changes in cytokine expression, dysbiosis, as well as links between the gut microbiome and other organs after RCI.

## 1. Introduction

The central health effect issue in many radiation exposure scenarios is whole-body ionizing radiation exposure combined with traumatic tissue injury (radiation combined injury, abbreviated RCI). It is estimated that 40–65% of the victims in Hiroshima and Nagasaki experienced physical trauma following ionizing radiation (IR) exposure [1,2]. In today’s world, nuclear power plant workers exposed to IR after a nuclear power plant accident or citizens living in or moving through contaminated areas often also experience physical injuries. The threat of RCI due to a nuclear or radionuclide-based terrorist explosive device is also a real-world possibility. It is reported that there are synergistic health effects caused by IR and traumatic tissue injury [1]. As a result, the complexity of the physiological responses to RCI remains poorly understood at the molecular, cellular, tissue, and organ levels, which are unique and more pronounced than either insult alone. In a nuclear accident, injuries suffered from nuclear explosions combine radiation injury (RI) with other forms of injury, such as burn, wound, hemorrhage, blast, trauma, and/or sepsis, severely increasing the risks of morbidity and mortality compared to RI alone [3]. To date, no U.S. FDA-approved countermeasures are available that specifically either prevent or treat RCI [1,4]. Therefore, this review covers RCI animal models, dynamic changes in cytokine expression, dysbiosis, as well as links between the gut microbiome and other organs after RCI.

## 2. Whole-Body Radiation Combined with Physical Trauma

A limited amount of research has been conducted to study survival after acute radiation syndrome (ARS) compared to RCI, and that conducted has exclusively involved experimental animals. In animal studies with mice [5,6,7,8,9,10,11,12,13], rats [14,15,16,17,18,19,20,21,22,23], guinea pigs [24], dogs [25,26], and swine [24], RCI with burning, wounding, and infection showed increased mortality after otherwise non-lethal radiation exposure [4]. Our laboratory has established experimental RCI mouse models with whole-body irradiation followed by hemorrhage, penetrating wound, burn wound, or bacterial infection to study survival [1]. RCI magnifies ARS [1]. RCI aggravated bone marrow cell depletion, peripheral blood cell count (CBC) depletion, spleen weight reduction, splenocyte reduction, gastrointestinal injury, brain bleeding, systemic bacterial infection, and tardied wound healing in comparison with RI [8,27,28,29].

### 2.1. Radiation Combined with Hemorrhage

#### 2.1.1. Whole-Body Radiation Combined with Hemorrhage

In the case of lethal whole-body ionizing radiation (IR, γ-photon at 8.75 Gy) followed by 20% hemorrhage (RCI with Hemo within 2 h post-IR), RCI with Hemo lowered WBCs (except elevations in neutrophils) at 4–5 h followed by a decrease that continued until day 3; the counts stayed at the nadir through day 15. RCI with Hemo lowered basophil (BAS), eosinophil (EOS), monocyte (MON), lymphocyte (LYM), and neutrophil (NEU) levels more than IR on day 1 or day 2. Unlike WBCs, RCI with Hemo decreased hematocrit, hemoglobin, and RBC levels on day 7 and day 15 more than IR, while Hemo alone returned to the basal level on day 7 and day 15. RBCs after RCI with Hemo were depleted faster than after IR. Hemo alone remarkably increased platelet counts on day 2, day 3, and day 7, and returned to the basal level on day 15. The data suggest that WBC depletion is a potential biomarker within the first 2 days post-IR and post-RCI with Hemo, whereas RBC depletion is a biomarker after 3 days post-IR and post-RCI with Hemo. At 4–5 h after RCI with Hemo but not Hemo alone, the NEU count increased, indicating exposure to RCI with Hemo. Counts of NEU at 4–5 h and platelets for day 2 through day 7 could be indicators for Hemo alone [30]. As for dynamic changes in cytokines and chemokines, RCI with Hemo excessively elevated IR-induced increases in G-CSF, CM-CSF, eotaxin, IFN-γ, MCP-1, MIP, TNF-α, IL-1β, IL-2, IL-3, IL-5, IL-6, IL-12, IL-13, IL-15, IL-17A, and IL-18 concentrations in serum from day 1 through day 15. In the ileum, RCI with Hemo significantly enhanced IR-induced IL-1β, IL-3, IL-6, IL-10, IL-12p70, IL-13, IL-18, and TNF-α concentrations, suggesting that increases in the expression of these cytokines and chemokines in the ileum of mice after RCI with Hemo contributed to increased serum concentrations [1].

#### 2.1.2. Whole-Body Radiation Combined with Extremity Trauma and Hemorrhage

Recently, a new rat model for a whole-body radiation combined with polytrauma was reported. Rats received RI (X-ray irradiator at 5.5 Gy) followed by a unilateral extremity trauma (fibular fracture + penetrating and crushed soft tissue injury), and then 90 min later received 37% step-wise Hemo. Either extremity trauma plus Hemo or RI alone resulted in 50% mortality, while RCI with extremity trauma plus Hemo displayed 80% mortality [31]. The increase in mortality was worse than for RCI with Hemo alone in our mouse model [32] due to (1) the additional fibular fractures and penetrating soft tissue injuries, and (2) differences between mice [32] and rats [31] used for the studies. No studies on rats receiving trauma before RI were reported, and no impact of this RCI on cytokines in this rat model was reported.

### 2.2. Whole-Body Radiation Combined with Skin Wound

In the case of whole-body ionizing radiation combined with skin wound trauma, there are several reports in mice [5,6,8,9,10,11,12,13].

Deoliveira et al. [9] used an ear punch model to study the effect of radiation on wound healing. A 2 mm diameter hole was made in each mouse ear using a clinical biopsy punch. Within 2 h after ear wounding, the mice were irradiated with X-ray at 3 Gy, 7 Gy, or 10 Gy. Healing occurred in a radiation dose-dependent manner. No study with ear wounding after IR was reported in this publication.

In our laboratory, mice received IR followed by a 15% total body surface area (TBSA) wound inflicted on the dorsal skin between two scapulae bones. We found that wound healing without IR took 14 days. By contrast, wound healing following IR was delayed up to 28 days [8]. Therefore, the healing rate was a potential indicator to separate IR alone, wounding alone, and RCI with wounding. IR resulted in a small healing bud [8], suggesting that delayed wound healing may be due to altered levels of proinflammatory chemokines and factors. RCI with wounding excessively magnified the cytokine storm, including G-CSF, eotaxin, MCP-1, MIP-1α, and MIP-1β, IL-1β, IL-6, IL-9, IL-10, IL-13, and KC, and increased iNOS expression due to RCI-induced upregulation of NF-κB p50, NF-κB p65, NF-IL6, NO production, nitrosative stress [8], and other signaling pathways in comparison with RI alone [33,34].

The timing of penetrating wound trauma and RI was a critical factor. When the wound was given prior to RI, RCI-induced mortality was reduced. However, when the wound was given after RI, RCI-induced mortality was enhanced [5,6,12]. Those mice receiving a penetrating wound prior to IR showed faster hematopoietic recovery. Wounded mice exhibited a splenic hypercellularity response that was approximately 10-fold higher than that of IR mice [35].

Brain hemorrhage was observed 12 days after RCI (RI with γ-photon at 9.5 Gy) with skin wound trauma (within 2 h post-RI) [36] or 3 days after RCI (RI with γ-photon at 9.5 Gy) with skin burn trauma (within 2 h post-RI) [37]. The change may be attributed to RCI-induced reduction in platelet counts [29]. However, RCI significantly elevated MIP-2, MCP-1, MIP-1α, KC, eotaxin, G-CSF, and IL-6 levels and vastly decreased MIG, IFN-γ, PDGF-bb IL-2, IL-9, and IL-10 levels in brain tissues more than RI alone [36].

RCI with skin wounding decreased gene expression of Cdh6 and increased gene expression of Timp3, Timp4, Itga7, mmp2, mmp3, mmp-9, mmp-10, mmp-13, Myd88, TLR1, TLR2, TLR3, TLR4, TLR6, TLR7, TLR8, and TLR9 more than RI alone on day 3 and day 7 post-RI [4]. Cdh6 is a cell–cell adhesion glycoprotein that is dependent on calcium. RCI-induced reduction of this glycoprotein was mediated by increases in Itga7 and mmps expression, even though Timps is thought to inhibit mmps and its expression was increased by RCI. The change in Timps expression may not have been sufficient to suppress mmps. Myd88 is a signal transducer involved in the activation of numerous proinflammatory genes. RCI-induced increases in Myd88 and mmps (which breaks down extracellular matrix) expression and reduction in Cdh6 expression are postulated to facilitate the serious bacterial infections that have been found in RCI mice. In addition, TLR expression on cell membranes was also significantly elevated. TLR4, whose binding ligands include the lipopolysaccharides of Gram-negative bacteria [38,39], was detected with a moderate increase in expression. RCI drastically increased TLR1, 2, 7, and 8 levels, which play crucial roles in the innate immune system. TLR1 and TLR2 are often detected on the cell surface and recognize bacterial components like lipopeptides. TLR7 and TLR8 are located intracellularly in endosomes and detect viral and bacterial nucleic acids, such as single-stranded RNA. Their activation triggers downstream signaling pathways. This activation leads to the production of inflammatory cytokines and other immune mediators to help the body fight off infections [40]. These results suggested that GI barrier dysfunction occurred and bacterial translocation from the intestinal lumen entered intestinal tissue. Subsequently, translocated bacterial entered other organs through the circulation, resulting in systemic infection, sepsis [8].

RCI with burning or wounding upregulated CRP [12] more than RI alone, whereas RCI with burning increased C3 and IL-18 levels in circulation more than RI alone [41]. C3 and IL-18 levels in blood were proportionally increased and positively correlated with each other. The C3 level was inversely correlated with the crypt depth of the ileum [42], whereas the IL-18 level was correlated with megakaryocyte abundance in bone marrow. A possible synergistic interaction between C3 inflammasomes and MAPK phosphorylation cannot be excluded. In the brain, C3 is known to be involved in neuronal synapse pruning by microglia [43,44] through the classical complement protein cascade [45], indicating that an increase in C3 levels but not IL-18 levels is detrimental to CNS synapses. Interestingly, AKT promoted DNA double-strand break (DSB) repair in cancer cells through the upregulation of Mre 11 expression following IR [46], indicating that in normal healthy tissues AKT may alter DSB repair after RCI. Thus, countermeasures that activate AKT are possibly beneficial to healthy cell survival [47].

Figure 1 depicts the possible mechanism underlying the molecular signaling pathways that are involved in proinflammatory cytokines and microRNAs leading to apoptosis, pyroptosis, and sepsis.

### 2.3. Whole-Body Radiation Combined with Skin Burning

In the case of whole-body ionizing radiation combined with skin burn trauma, there are reports using mice [1,48], rats [14,17,19,20,21,22,23], guinea pigs [24], dogs [25,26], and swine [24]. In our laboratory, 15% TBSA skin burning was conducted with a 1 × 1 inch custom-designed template, which was positioned centrally over the shaved dorsal skin surface. Within 10 min to 48 h before or after IR, ethanol (95%) with a volume of 0.25 mL was evenly applied to the dorsal skin surface exposed by the template. The ethanol was ignited and allowed to burn for 12 s after IR (γ-photon at 9.75 Gy) [12]. When mice received skin burns after IR, mortality increased by 14–20%, but the increase in mortality was independent of the time interval between IR and burning. When burning occurred before IR, mortality was only 0–8%. Although cytokine/chemokine levels in blood were not measured in these mice, corticosterone, C-reactive protein (CRP), complement protein 3 (C3), immunoglobulin M (IgM), and prostaglandin E_2_ (PGE_2_) levels were measured. RCI with burning showed no synergistic effects on corticosterone but synergistically increased CRP, C3, and PGE_2_ levels and decreased IgM levels, while IR alone significantly increased corticosterone, CRP, C3, IgM, and PGE_2_ levels, and burning alone significantly increased corticosterone, C3, IgM, and PGE_2_ levels but decreased CRP levels [12].

In another laboratory, a copper rod was heated in 100 °C boiling water and applied to the mouse dorsum and flank for 10 s. This application resulted in a 20% TBSA burn. These mice immediately received IR (X-ray beam at 2–9 Gy) [48]. In this case, burn injury followed by IR increased mortality in an IR dose-dependent manner. RCI with burning significantly increased levels of pro-inflammatory cytokines IL-1β, IL-6, IL-10, and IL-12 more than RI alone. Immature myeloid cells were also induced in these mice. This laboratory did not study burns received after IR. Therefore, the severity of mortality between burns given before or after IR could not be compared [48]. More information on innate immunity after RCI can be found in the review published by Kumar et al., 2025 [41].

The key feature of RCI with burning is extensive and severe GI injury, resulting in GI dysfunction in absorption, secretion, and tight junction, thus increasing the risk of infection. Delayed burn wound healing often occurred and mortality was observed early post-RCI [15]. In rat serum, IL-1β, IL-6, and TNF-α levels were significantly elevated [49,50,51], and increased internal bacterial infection and toxins were detected [52], implicating internal bacterial infection and toxin release as major factors in increased mortality after RCI [52].

Like penetrating wounds, the timing of burning and IR in mice is a critical factor. When burning was received prior to irradiation, IR-induced mortality was reduced. By contrast, when burning was received after IR, IR-induced mortality was increased [12]. However, in dogs, similar mortality was observed regardless of receiving burn injury before or after IR [25].

### 2.4. Whole-Body Radiation Combined with Bacterial Exposure

#### 2.4.1. In Vivo Studies

During in vivo studies of whole-body radiation combined with bacterial exposure, B6D2F1 female mice received ^60^Co γ-photon at 8 Gy followed by oral inoculation with a dose of 10^8^ *Klebsiella pneumoniae* administered 4 days after IR. On day 30 post IR, 15% of mice survived [53,54]. Treatment with ofloxacin post-RCI with bacteria for a 7-day consecutive treatment or a 21-day consecutive treatment was effective, with 55% 30-day survival and 90% 30-day survival, respectively [54]. Similarly, RCI with *Pseudomonas aeruginosa* sepsis in mice increased mortality on day 30 post-RCI [55]. When these mice received ^60^Co γ irradiation at 6 Gy and *B. anthracis* Steme infection (4.4 × 10^8^ CFU administered intratracheally), a synergistic effect on 30-day mortality (6 Gy: 0%; *B. anthracis*: 25%; 6 Gy + *B. anthracis*: 100%) was observed [56].

In another animal model, male Sprague-Dawley rats received 5 Gy radiation. Forty-eight hours later, these rats received cecal ligation and puncture (CLP) [16]. This RCI with CLP resulted in 62% mortality from day 4 through day 10. Their serum IL-6 and TNF-α levels were significantly increased, and myeloperoxidase activity in the kidneys, lungs, and small intestine was vastly elevated compared to RI alone.

Another experimental model was developed to study weightlessness and irradiation [57,58]. Mice received hindlimb suspension and solar particle event (SPE)-like radiation (^60^Co or proton beam at 1–2 Gy). Five days later, these mice were challenged with *Klebsiella pneumoniae* and *Pseudomonas aeruginosa* through systemic and pulmonary routes. Clearance of bacterial infection was found to be impaired in these mice, compared to clearance in blood or tissues 5 days after bacterial exposure in control mice [59], indicating that an astronaut during extended space travel would be at increased risk for developing infections. No IR followed by hindlimb suspension was reported in this study.

#### 2.4.2. In Vitro Studies

During in vitro studies of mesenchymal stem cells (MSCs) that were exposed to IR (γ-photon at 8 Gy) followed by *E. coli* or *S. epidermidis* (5 × 10^7^ bacteria/mL) for 3 h, both IR and RCI with bacteria on day 1 post-RI increased p62/SQSM1, SUMO1, collagen III, LC3, Sirt, mmp13, and mmp3 levels more than RI alone [60]. MSCs manifested their capability of phagocytizing *E. coli* or *S. epidermidis* [60]. The results indicated the potential role of MSCs in sustaining the antibacterial barrier function of irradiated tissues. It is postulated that effector mechanisms expressed by MSCs enable them to contribute to the innate defense response after RI alone or, especially, RCI with bacteria, burning, wounding, or even Hemo.

More recently, 16S rRNA expression was measured using gene targeted primers (forward (5′-ACTCCTACGGGAGGCAGCAGT-3′) and reverse (5′-TATTACCGCGGCTGCTGGC-3′) with the QuantStudio 3 Realtime PCR System. PCR-derived bacterial counts were expressed as nanograms of bacterial DNA per gram of mouse organ tissue [61,62]. This was used to measure the presence and concentrations of bacteria. Bacterial 16S rRNA gene expression in tissues of RCI mice was higher than that in tissues of RI mice because of the intestinal barrier dysfunction caused by IR [63]. In our laboratory, we also measured LPS levels to measure the presence and concentration of Gram-negative bacteria [63].

Taken together, Table 1 lists different types of RCI, and Table 2 lists cytokines/chemokines that exhibit dynamic changes under different types of RCI.

## 3. Impacts of Radiation Combined with Skin Wounding on the Microbiome

Most organ tissues from healthy animals are sterile (except for occasional, transient bacteremia), so the presence of bacteria in detectable numbers is indicative of systemic infection. This laboratory aseptically cultured heart blood and homogenized liver tissue from sham-treated, wounded, irradiated, or combined injury mice to determine whether facultative bacteria had entered the circulation. In sham-treated mice, no bacteria were found, whereas in wounded and irradiated mice, *Enterococcus* sp. and *Staphylococcus* sp. were occasionally found. In mice with RCI (RI with γ-photon at 9.5 Gy followed by skin wounding within 2 h post-RI), *Enterococcus* sp., *Staphylococcus* sp., *Bacillus * sp., and *Lactobacillus* sp. were found in tissues. *Enterococcus* sp., *Staphylococcus* sp., *Bacillus* sp., and *Lactobacillus* sp. were also isolated from the ileum of these RCI mice. Bacteremia in wounded mice was transient and was seen only on the third day after wounding. Systemic infection was demonstrated in RCI mice through day 17 and sporadically in irradiated mice through day 25. In RCI mice, *Bacillus* and *Lactobacillus* were isolated within the first 8 days after RCI [8]. The results suggested that these bacteria entered through the wound site. These data implied that wounded mice resisted infection from the ileum, while systemic infection in RCI mice was accelerated by several days compared to infection in irradiated mice, after which death occurred.

Horseman et al. [62] reported that whole-body radiation with γ-photon at 9.5 Gy caused no significant alpha diversity differences, while beta diversity shifts and taxonomic profiles revealed significant alterations in *Akkermansia*, *Bacteroides*, and *Lactobacillus*. This study provides a framework for the identification of microbial elements that may influence radiosensitivity, biodosimetry, and the efficacy of potential therapeutics. No RCI was reported in this study.

Mitra et al. [64] reported that in C57BL/6 male mice, partial-body irradiation with a leg out at 13 Gy (with x-ray beam) revealed a reduction in the number of beneficial bacteria, such as *Alistipes*, *Eubacterium*, *Lactobacillus*, and *Bifidobacterium*, and an increase in the number of potentially pathogenic bacteria, such as *Enterococcus* and *Staphylococcus*, in both luminal and mucosal samples of irradiated animals along with increases in TNFα and KC levels compared with unirradiated animals. No RCI was reported in this study.

## 4. Gut Microbiota Links to Other Organs

The microbiome can be divided into the oral microbiome, skin microbiome, lung microbiome, gut microbiome, nasopharyngeal microbiome, urogenital microbiome, and ocular/lacrimal microbiome [65]. Different microbiota comprise different types and abundances of microbial species, thereby maintaining the homeostatic condition of each body area. Species of importance include the phyla Firmicutes, Bacteroidetes, Actinobacteria, and Proteobacteria in the GI [66,67,68]; *Veillonella* sp., *Actinomyces* sp., *Neisseria* sp., *Simonsiella* sp., and *Eubacterium* sp. in the oral cavity [69,70]; *Staphylococcus epidermidis*, *Micrococcus luteus*, and *Staphylococcus aureus* on skin [71]; *Prevotella* sp. and *Veillonella* sp. in the lung [72,73,74]; *Propionibacterium acnes*, *Staphylococcus epidermidis*, and *Corynebacterium tuberculostearicum* in the naso-pharyngeal area [75]; and *Lactobacillus iners*, *Lactobacillus crispatus*, *Lactobacillus gasseri*, and *Lactobacillus jensenii* in the vaginal area [76,77,78]. There are extensive and complex interactions across the distinct microbial communities spanning the body, including the so-called gut–lung axis [79,80,81,82,83] and the microbiota–gut–brain axis [84,85,86,87,88,89,90,91,92]. Some immune cells can discriminate between pathogenic and commensal bacteria [93,94,95]. It is reported that the intestinal microbiota acts as a protective regulator against radiation pneumonitis [82]. A population of mice was reported to recover from high-dose radiation to live normal life spans [95]. These “elite survivors” harbored a distinct gut microbiota that developed after radiation and protected against radiation-induced damage and death in both germ-free and conventionally housed recipients. Elevated abundances of members of the bacterial taxa *Lachnospiraceae* sp. and *Enterococcaceae* sp. were found to be associated with post-radiation restoration of hematopoiesis and gastrointestinal repair. These bacteria were also found to be more abundant in leukemia patients undergoing radiotherapy who displayed milder gastrointestinal dysfunction [83]. Metabolomics analysis revealed increased fecal concentrations of microbially derived propionate and tryptophan metabolites in elite survivors. The administration of these metabolites caused long-term radioprotection, mitigation of hematopoietic and gastrointestinal syndromes, and a reduction in proinflammatory responses [95].

Whole-body irradiation RCI can cause changes in the skin microbiome and gut microbiome [8]. It is unclear whether RCI also altered the oral microbiome, lung microbiome, nasopharyngeal microbiome, urogenital microbiome, or ocular/lacrimal microbiome.

Dysbiosis refers to an imbalance among microbial communities (like bacteria, fungi, viruses, etc.) within the body, often specifically in the gut. This imbalance can manifest as a decrease in commensal microbes, an increase in pathogenic microbes, or a loss of overall microbial diversity. Dysbiosis is linked to various health problems, including digestive issues, inflammation, and even some systemic diseases. Whole-body radiation-induced dysbiosis of the gut microbiota was associated with the progression of radiation-induced intestinal injury [96] but also affected other organs.

Gut dysbiosis not only precipitates digestive tract diseases such as inflammatory bowel disease but is also is associated with chronic obstructive pulmonary disease and asthma [97]. Additionally, evidence has proven that the gut microbiome is involved in cognitive dysfunction. Quercetin inclusion complex gels ameliorated radiation-induced brain injury by regulating the gut microbiota [98]. Changes in the gut microbiome have also been reported to cause atherosclerosis [99] or other cardiovascular diseases [100,101].

The GI microbiota can communicate with the brain (gut–brain axis) via various pathways and molecules, such as the enteric nervous system, the vagus nerve, microbial metabolites, and the immune system [102]. Alterations in the composition of the GI microbiome can lead to alterations in its functional metabolic output and means of communication, therefore potentially causing downstream cognitive effects [84]. Similar communications between the gut microbiota with the kidney [103] or skin diseases [104,105] have been demonstrated. Taken together, the gut–brain axis, gut–lung axis, gut–heart axis, gut–kidney axis, and gut–skin axis are present in the literature. It will not be surprising to see more reports on gut microbiota axes with the other organs, including bone and the pancreas, spleen, and liver. Consequently, studying how radiation can affect this important network of communication could lead to new and critical interventions, as well as prevention strategies. For example, it is reported that gut-derived probiotics [96], valeric acid [106], L-histidine [107], or fecal extracts either protected against or ameliorated radiation pneumonia [82]. Whether sex is a confounding factor in RCI-induced dysbiosis remains unclear and needs further exploration.

## 5. A Biomolecule Panel for Estimation of Radiation Dose

It is important to estimate the radiation dose in the case of an unexpected nuclear accident. There is no microbiota available as a biomarker for the estimation of radiation dose yet, even though animal bacterial microbiome data could be used to predict an animal’s radiation status. The fungal microbiome exhibited no significant differences regarding genotype or time after radiation exposure [108]. The level of Flt-3 ligand in blood (a biomarker of bone marrow aplasia) was increased in a radiation dose-dependent manner [1,109]. Wounding did not change the IR-induced increase in Flt-3 ligand levels after IR + wounding [1]. Likewise, the IR- or IR + wound-induced dynamic changes in CD27 (receptors on the lymphocyte surface) and miR-34a (an inhibitor of Bcl-2) levels were similar [1], suggesting that Flt-3 ligand, CD27, and miR-34a may be a useful panel that can be used to estimate radiation dose regardless of IR alone or combined with additional trauma.

It is evident that gene regulation can be used as a potential biomarker in this regard. In human peripheral blood after irradiation, weighted gene co-expression network analysis (WGCNA) and differentially expressed gene (DEG) co-analysis of RNA-sequencing data from the Gene Expression Omnibus (GEO) database identified seven radiation-induced specific genes, with two downregulated genes and five upregulated genes. Those five radiation-specific genes (CCNG1, CDKN1A, GADD45A, GZMB, and PHLDA3) showed a strong linear correlation with the total-body X-ray radiation model. Among them, the CCNG1 and CDKN1A genes best fit the radiation dose–response relationship across both mice and humans based on receiving operator characteristic (ROC) curve analysis. Moreover, the CCNG1 protein could accurately predict the absorbed dose for up to 28 days after radiation exposure (>95%). The data suggested that CCNG1 mRNA and CDKN1A mRNA were optimal in predicting the radiation dose response, regardless of trauma, burn, age, and sex. Moreover, the CCNG1 protein showed a strong linear correlation that was positively associated with radiation dose and time post-IR. CCNG1 protein levels remained increased up to day 28 [110], suggesting this protein as a leading biomarker for assessing radiation dose.

## 6. Drugs for Treating Radiation Combined Injury

There are multiple underlying mechanisms involved in RCI. Due to the mechanistic complexity, drugs or remedies are yet to be fully defined [1]. In the past 18 years, many attempts to do so have been undertaken in this lab. We have shown efficacy with treatment with Alxn4100TPO, a TPO receptor agonist, via increasing platelets [111]; bone marrow transplant [112]; mesenchymal stem cells [113]; ghrelin by reducing IL-1ß, IL-6, IL-17A, IL-18, KC, and TNF-α levels in serum but sustaining G-CSF, KC, and MIP-1α increases in the ileum [1,63]; ciprofloxacin by increasing IL-3 levels and RBCs [114]; WR-151327 [112]; silvadene [112]; and trichostatin by reducing IL-1β, IL-6, CRP and TNF-α levels [18], resulting in increased survival after RCI. The underlying mechanisms involve the inhibition of miR-34a by G-CSF in specific tissues and circulation, several signaling pathways, including increases in AKT activation and decreases in MAPK activation, and reduction of apoptosis as well as pyroptosis [42]. Moreover, in the hippocampus of the brain, brain-derived neurotrophic factor (BDNF) was downregulated via increased miR-34a in the small intestine and peripheral blood in mice with total abdominal irradiation, resulting in cognitive dysfunction. The miR-34a–BDNF axis was validated through miR-34a antagomir injection [90].

Combinational therapies of S-TDCM plus gentamicin [112], neulasta plus Alxn4100TPO [115], or neulasta plus citrulline [116] are also effective for enhancing survival after RCI, likely by enhancing survival of hematopoietic stem/progenitor cells, gastrointestinal repair, or accelerating recovery of cutaneous wounds. The mechanism of this enhanced survival benefit is currently under investigation.

The effects of the microbiome on the radiation response are known to be modulated by targeted treatment with antibiotics, probiotics [117], diet, prebiotics, vitamin, minerals [65], and fecal microbiota transplant (FMT) [107,118]. Nevertheless, limited studies on RCI are reported. Levofloxacin and amoxicillin administered together once daily for 21 days after RCI did not improve survival up to 30 days [11].

Table 3 summarizes the drugs that are effective in treating RCI.

RCI research has also contributed to recent investigations regarding the human virus SARS-CoV-2 and the related disease COVID-19. Prospectively, RCI mitigation studies could suggest treatments to blunt the cytokine storm associated with COVID-19 [119,120] because of the similarity of cytokines affected either by RCI or COVID-19 [120]. COVID-19 increases levels of angiotensin II, stimulating the synthesis of not only proinflammatory cytokines TNF-α, IFN-γ, IL-6, and IL-1β [121] but also anti-inflammatory cytokines IL-10 and TGF-β1, which may induce M2 macrophage polarization [122] and avoid activation of the γδ T lymphocytes required to pledge the antiviral immune response [123]. The possibility that cytokines in the blood and/or tissues/organs regulate MAPK activation cannot be excluded and should be explored. Nevertheless, medical countermeasures developed to treat RCI might be relevant in treating COVID-19, particularly COVID-19-induced pulmonary edema, as well as other emerging infectious diseases that result in a cytokine storm. Again, antibiotics, probiotics, diet, prebiotics, vitamin, minerals, and FMT that have been studied to regulate the gut microbiome after whole-body RCI could be useful for protection against COVID-19 infection. It is important to bear in mind that COVID-19 vaccine side effects occur [124,125], and RCI countermeasures as adjuvants might possibly alleviate these side effects.

## 7. Conclusions

This report provides a comprehensive description of the complicated nature of radiation injury followed by trauma such as hemorrhage, wound, burn, or infection. RCI induces greater mortality, delays wound healing, and causes an excessive cytokine storm, inflammation, and bacterial sepsis, which are mediated by biomolecular dynamic changes at the molecular, cellular, tissue, and organ levels (Figure 2). As one approach to save victims after radiological accidents or nuclear events, prevention or mitigation of the gut microbiota is critical for saving lives through the modulation of molecules involved in signal transduction pathways. Furthermore, studying how radiation can affect this important network of communication between the gut microbiota and other major organs, such as the brain, lung, heart, kidney, and skin, could lead to new and critical interventions, as well as prevention strategies.

## Figures and Tables

**Figure 1 ijms-26-10456-f001:**
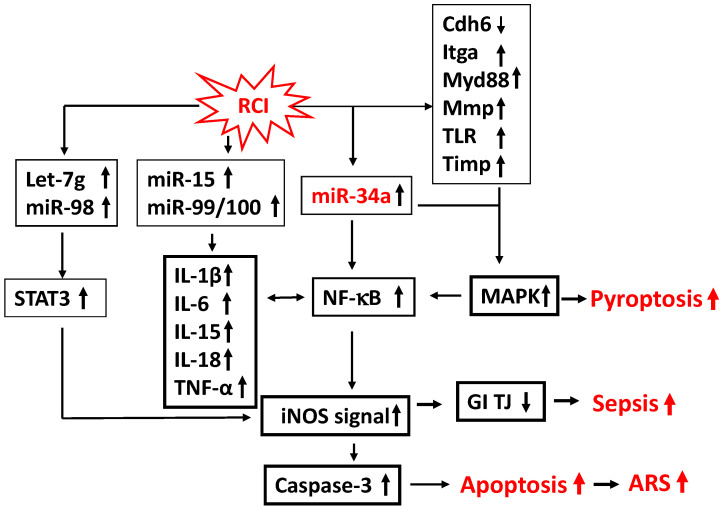
A possible mechanism underlying cell death and mortality in the ileum of mice exposed to RCI (radiation followed by wound trauma) [4,8]. Up arrow: increase; down arrow: decrease.

**Figure 2 ijms-26-10456-f002:**
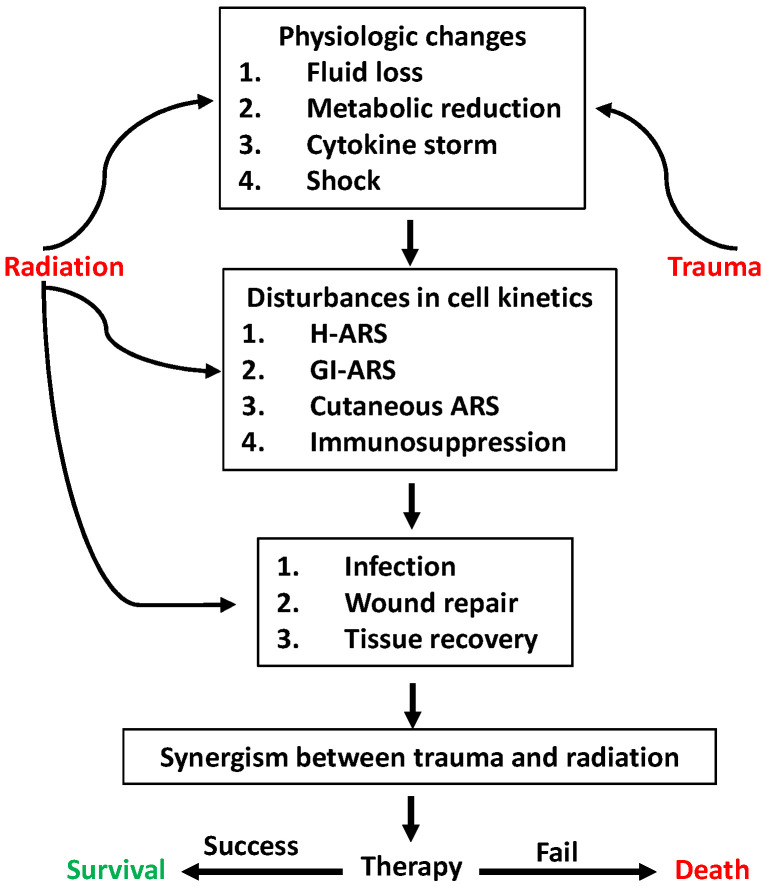
Radiation and trauma synergistically impact morbidity and mortality through various actions. Trauma and radiation cause physiologic changes. However, radiation but not trauma induces H-ARS, GI-ARS, cutaneous-ARS, and immunosuppression. Then, infection, wound repair, and tissue recovery are impacted. Subsequently, synergistic effects between trauma and radiation appear. Effective therapy given in time can save victims. H-ARS: hematopoietic acute radiation syndrome; GI-ARS: gastrointestinal acute radiation syndrome.

**Table 1 ijms-26-10456-t001:** Different types of radiation combined injury (RCI).

Type	Species	References
Rad + 20% Hemo	Mouse	[30]
Rad + 37% Hemo + Penetrating soft tissue injury	Rat	[32]
Rad + Skin wound	Mouse	[8]
Rad + Ear punch wound	Mouse	[9]
Rad + Skin burn	Mouse	[12,48]
	Rat	[14,17,19,20,21,22,23]
	Guinea pig	[24]
	Dog	[25,26]
	Swine	[24]
Rad + 15% total body surface area burn	Mouse	[12]
Rad + 20% total body surface area burn	Mouse	[48]
Rad + Infection	Mouse	[53,54,55,56]
Rad + CLP	Rat	[16]
Rad+ hindlimb suspension + bacteria	Mouse	[59]

Rad: radiation; CLP: cecal ligation punch.

**Table 2 ijms-26-10456-t002:** Cytokines/chemokines affected after different types of radiation combined injury (RCI).

Type	Cytokine/Chemokine	Species	References
Rad + Wound	IL-1β, IL-6, IL-9, IL-10, IL-13,	Mouse	[8,41]
	IL-18, KC, G-CSF, eotaxin,		
	MCP-1, MIP-1α, MIP-1β CRP, C3		
Rad + Skin burn	IL-1β, IL-6, IL-10, IL-12, TNF-α	Rat	[49,50,51]
Rad + CLP	IL-6, TNF-α	Rat	[16]
Rad + Infection	KC, TNF-α	Mouse	[64]

Rad: radiation; CLP: cecal ligation punch.

**Table 3 ijms-26-10456-t003:** Drugs for treating radiation combined injury.

Drug	Actions	Species	References
Alxn4100TPO	TPO receptor agonist	Mouse	[111]
	Increased platelets		
BM transplant	Increased bone marrow cells	Mouse	[112]
Bone marrow MSCs	Increased bone marrow cells	Mouse	[113]
Ghrelin	Decreased IL-1β, IL-6, IL-17A	Mouse	[63]
	IL-18 levels		
	Increased G-CSF, KC, MIP-1α levels		
	Vagotomy	Rat	[16]
Ciprofloxacin	Increased IL-3 levels and RBCs	Mouse	[114]
WR-151327	Not identified	Mouse	[112]
Silvadene	Not identified	Mouse	[112]
Trichostatin	Increased IL-1β, IL-6, CRP	Rat	[18]
	TNF-α levels		
S-TDCM + Gentamicin	Not identified	Mouse	[112]
Neulasta + Alexn4100TPO	Improved H-ARS and GI-ARS	Mouse	[115]
Neulasta + Citrulline	Improved endothelium	Mouse	[116]

TPO: Thrombopoetin; MSC: Mesenchymal stem cells; WR-151327: S-3-(3-methylaminopropylamino) propylthiophosphorothioic acid; S-TDCM: Synthetic trehalose dicorynomycolate.

## Data Availability

No new data were created or analyzed in this study. Data sharing is not applicable to this article.

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
