# Peer review of "An Update on Dynamic Changes in Cytokine Expression and Dysbiosis Due to Radiation Combined Injury"

_ijms, 2025, doi:10.3390/ijms262110456_

Round 1

Reviewer 1 Report

Comments and Suggestions for Authors

This article entitled “An Update on Dynamic Changes in Cytokine Expression and Dysbiosis due to Radiation Combined Injury” reviewed the changing patten of cytokine and gut microbiota after RCI. The overall structure of the manuscript is clear and coherent. The manuscript addresses an important and timely topic and provides a broad overview of the field. However, there are some limitations that need to be addressed.

  1. There is no introduction about the changes of cytokines in whole-body radiation combined with hemorrhage and extremity trauma, please supplement the changes in cytokines, which is the theme of this review.
  2. It is suggested to add a summary diagram of signaling pathways in whole-body radiation combined with skin wound.
  3. In line 152, a reference numbered 12 was cited, and preparation methods of animal model with whole-body radiation combined with skin burn is induced, however, the author did not disclose the results of this study, which is more important information for the readers.
  4. The section of 7, COVID-19 and cytokines, seems redundant. It is not related to RCI.

Once the above concerns are fully addressed, the manuscript could be accepted for publication in this journal.

Author Response

Reviewer 1

Comments from the reviewer 1

This article entitled “An Update on Dynamic Changes in Cytokine Expression and Dysbiosis due to Radiation Combined Injury” reviewed the changing patten of cytokine and gut microbiota after RCI. The overall structure of the manuscript is clear and coherent. The manuscript addresses an important and timely topic and provides a broad overview of the field. However, there are some limitations that need to be addressed.

  1. There is no introduction about the changes of cytokines in whole-body radiation combined with hemorrhage and extremity trauma, please supplement the changes in cytokines, which is the theme of this review.
  2. It is suggested to add a summary diagram of signaling pathways in whole-body radiation combined with skin wound.
  3. In line 152, a reference numbered 12 was cited, and preparation methods of animal model with whole-body radiation combined with skin burn is induced, however, the author did not disclose the results of this study, which is more important information for the readers.
  4. The section of 7, COVID-19 and cytokines, seems redundant. It is not related to RCI.

Once the above concerns are fully addressed, the manuscript could be accepted for publication in this journal.

Author's responses:

Thank you for this reviewer’s constructive comments and suggestions that have been incorporated into the manuscript.

Comment 1. There is no introduction about the changes of cytokines in whole-body radiation combined with hemorrhage and extremity trauma, please supplement the changes in cytokines, which is the theme of this review.

Response: I think you have meant RCI with hemorrhage. The dynamic changes in cytokines/chemokines are inserted. Please see lines 77-83. As for RCI with hemorrhage and extremity trauma, the publication does not include the cytokine data. Therefore, no such data can be included in this review manuscript. Please see lines 94 and 95.

Comment 2. It is suggested to add a summary diagram of signaling pathways in whole-body radiation combined with skin wound.

Response: Thank you very much for making this suggestion. Please see Figure 1 and lines 162-167.

Comment 3. In line 152, a reference numbered 12 was cited, and preparation methods of animal model with whole-body radiation combined with skin burn is induced, however, the author did not disclose the results of this study, which is more important information for the readers.

Response: Sorry for our being careless. Now the results of this study are included. Please see lines 177-185.

Comment 4. The section of 7, COVID-19 and cytokines, seems redundant. It is not related to RCI.

Response: The section of 7 with COVID-19 and cytokines has been revised and merged into the section of 6. The reasons for including this discussion are (a) COVID-19 is still ongoing and patients died every day worldly and international wide; (b) there is a similarity between cytokine storm caused by Covid-19 and RCI; and (c) COVID-9 vaccines have side-effects that can be lethal. Therefore, I think an insightful discussion is inspiring to the readers. The section of 7 in the revised manuscript is Conclusion instead. Please see lines 449-465.

--- All revised/changed areas are highlighted in yellow for the reviewer to review and evaluated.

Reviewer 2 Report

Comments and Suggestions for Authors

Summary

This manuscript is aimed at reviewing research on the combined effects of ionizing radiation with other forms of injuries, such as burn, wound, hemorrhage, blast, trauma, and/or sepsis, a situation that can be common in a nuclear accident. The authors specifically address the cytokine expression and dysbiosis, mainly in animal models.

General Comments

This manuscript can be valuable as it describes the complex nature of the radiation combined injuries (RCI), addressing several aspects of them, such as possible mechanisms, possible biomarkers for radiation dose, and possible drugs for treating these effects.

However, it should be revised since:

  • The authors have to make clear what is new in this review compared to their previous article (IJRB 2023), taking into account that the title (“An update….”) has to be reflected in the content.
  • The abstract needs to be reorganized since it includes a paragraph that recounts specific research, which is not suitable for a review article (see specific comments).
  • In the text, the authors do not always specify the sequence of RCI, i.e., whether radiation comes before or after the other injuries and when. The time window between radiation and wounding, etc, may affect the response, as the authors themselves mention in several cases.
  • In several cases, the authors describe the RCI without comparison with the RI alone (see e.g., sect. 2.2). This comparison is essential for understanding the interaction between radiation and other injuries, so that the usefulness of these data for the present paper should be motivated.
  • Not always are the relevant conditions mentioned, such as the doses used.

Specific comments

  • Abstract, lines 15-21. The paragraph is an account of research carried out in the authors' lab, with some details that are not suitable for an abstract of a review paper.
  • Lines 39-40. What is the evidence for a synergy between radiation and trauma ? Some references are appropriate here,
  • Line 196. Please specify if SPE-like radiation is a proton beam.
  • Line 219, Tables 1 and 2. No reason why they are presented in a section dealing with radiation plus bacterial exposure. They should be moved to a more suitable position in the text.
  • Lines 361-62. Explain. Not clear what the correlation is (radiation dose correlates with time post-irradiation ?), even if it is taken from the abstract of the cited paper.
  • Line 415. It would be better to say “SARS-CoV-2 and the related disease COVID-19”.
  • Line 443. Fig.1. Some aspects of the flowchart need to be clarified/explained, such as: the mechanisms by which radiation and trauma cause indirect effects; why the “Immunosuppression” is placed as the consequence of “disturbance in cell kinetics” (the two boxes are connected with an arrow) while both can be induced by radiation (directly) and trauma (indirectly).
  • Line 460. Acknowledgments. The section bears a resemblance to a report intended for domestic circulation. For example, the locations for people, Departments, etc. are not specified, which would ensure complete information for a non-US reader.

Author Response

Reviewer 2

Comments and Suggestions for Authors

Summary

This manuscript is aimed at reviewing research on the combined effects of ionizing radiation with other forms of injuries, such as burn, wound, hemorrhage, blast, trauma, and/or sepsis, a situation that can be common in a nuclear accident. The authors specifically address the cytokine expression and dysbiosis, mainly in animal models.

General Comments

This manuscript can be valuable as it describes the complex nature of the radiation combined injuries (RCI), addressing several aspects of them, such as possible mechanisms, possible biomarkers for radiation dose, and possible drugs for treating these effects.

However, it should be revised since:

  • The authors have to make clear what is new in this review compared to their previous article (IJRB 2023), taking into account that the title (“An update….”) has to be reflected in the content.
  • The abstract needs to be reorganized since it includes a paragraph that recounts specific research, which is not suitable for a review article (see specific comments).
  • In the text, the authors do not always specify the sequence of RCI, i.e., whether radiation comes before or after the other injuries and when. The time window between radiation and wounding, etc, may affect the response, as the authors themselves mention in several cases.
  • In several cases, the authors describe the RCI without comparison with the RI alone (see e.g., sect. 2.2). This comparison is essential for understanding the interaction between radiation and other injuries, so that the usefulness of these data for the present paper should be motivated.
  • Not always are the relevant conditions mentioned, such as the doses used.

Specific comments

  • Abstract, lines 15-21. The paragraph is an account of research carried out in the authors' lab, with some details that are not suitable for an abstract of a review paper.
  • Lines 39-40. What is the evidence for a synergy between radiation and trauma ? Some references are appropriate here,
  • Line 196. Please specify if SPE-like radiation is a proton beam.
  • Line 219, Tables 1 and 2. No reason why they are presented in a section dealing with radiation plus bacterial exposure. They should be moved to a more suitable position in the text.
  • Lines 361-62. Explain. Not clear what the correlation is (radiation dose correlates with time post-irradiation ?), even if it is taken from the abstract of the cited paper.
  • Line 415. It would be better to say “SARS-CoV-2 and the related disease COVID19”.
  • Line 443. Fig.1. Some aspects of the flowchart need to be clarified/explained, such as: the mechanisms by which radiation and trauma cause indirect effects; why the 2 “Immunosuppression” is placed as the consequence of “disturbance in cell kinetics” (the two boxes are connected with an arrow) while both can be induced by radiation (directly) and trauma (indirectly).
  • Line 460. Acknowledgments. The section bears a resemblance to a report intended for domestic circulation. For example, the locations for people, Departments, etc. are not specified, which would ensure complete information for a non-US reader.

Authors’ responses

We greatly appreciated reviewer 2’s comments and suggestions that have incorporated into the revised manuscript. The modified areas are highlighted for you to review.

  • The authors have to make clear what is new in this review compared to their previous article (IJRB 2023), taking into account that the title (“An update….”) has to be reflected in the content.

Response: Yes, we had done that with sentences inserted in both the Abstract and Introduction. Please see lines 25-27 and 47-49.

  • The abstract needs to be reorganized since it includes a paragraph that recounts specific research, which is not suitable for a review article (see specific comments).

Response: Yes, we had revised the Abstract accordingly. Please see lines 15-17 and 25-27.

  • In the text, the authors do not always specify the sequence of RCI, i.e., whether radiation comes before or after the other injuries and when. The time window between radiation and wounding, etc, may affect the response, as the authors themselves mention in several cases.

Response: Yes, we had inserted the information per your comments and suggestions. They are scattered through the entire manuscript with highlighted areas.

  • In several cases, the authors describe the RCI without comparison with the RI alone (see e.g., sect. 2.2). This comparison is essential for understanding the interaction between radiation and other injuries, so that the usefulness of these data for the present paper should be motivated.

Response: Thank you for pointing it out. The RCI data were compared with the RI data. Therefore, those words were added as needed. They are scattered through the entire manuscript with highlighted areas.

  • Not always are the relevant conditions mentioned, such as the doses used.

Response: The doses that were studied are included. They are scattered through the entire manuscript with highlighted areas.

Specific comments

  • Abstract, lines 15-21. The paragraph is an account of research carried out in the authors' lab, with some details that are not suitable for an abstract of a review paper.

Response: The abstract has revised accordingly. Please see lines 15-17 and 25-26.

  • Lines 39-40. What is the evidence for a synergy between radiation and trauma? Some references are appropriate here,

Response: The sentence has been revised accordingly. Please see lines 39-40.

  • Line 196. Please specify if SPE-like radiation is a proton beam.

Response: Yes, it was cobalt or the proton beam as the article indicated. The information has been inserted. Please see line 226.

  • Line 219, Tables 1 and 2. No reason why they are presented in a section dealing with radiation plus bacterial exposure. They should be moved to a more suitable position in the text.

Response: Thanks. We inserted these two tables in a section dealing with radiation plus bacterial exposure, because Table 1 includes animal models of Rad+infection, Rad+CLP, and Rad+hindlimb suspension+bacteria (lines 266-268) and Table 2 includes cytokine data after both Rad+CLP and Rad+infection (lines 278 and 279). Therefore, we think it will be fine to present these two tables here if it is fine with you.

  • Lines 361-62. Explain. Not clear what the correlation is (radiation dose correlates with time post-irradiation?), even if it is taken from the abstract of the cited paper.

Response: The correlation was positively associated with radiation doses and times post-IR. The CCNG1 protein stayed increases up to day 28, suggesting this protein a leading biomarker for assessing radiation doses. Please see lines 395-397.

  • Line 415. It would be better to say “SARS-CoV-2 and the related disease COVID19”.

Response: Thank you for your suggestion. We modified it accordingly. Please see line 450.

  • Line 443. Fig.1. Some aspects of the flowchart need to be clarified/explained, such as: the mechanisms by which radiation and trauma cause indirect effects; why the 2 “Immunosuppression” is placed as the consequence of “disturbance in cell kinetics” (the two boxes are connected with an arrow) while both can be induced by radiation (directly) and trauma (indirectly).

Response: Thank you for pointing out the mistakes on Fig.1. (Fig. 2 instead in the revised version) We corrected/clarified it.  Please see the revised Fig. 2 and its legend (please see lines 476-483).

  • Line 460. Acknowledgments. The section bears a resemblance to a report intended for domestic circulation. For example, the locations for people, Departments, etc. are not specified, which would ensure complete information for a non-US reader.

Response: Thank you for pointing out the deficiency. We inserted required information to the section.  Please see the lines 496-502.

Reviewer 3 Report

Comments and Suggestions for Authors

Review Comments

This review provides a comprehensive description of the complicated nature of radiation injury followed by trauma such as hemorrhage, wound, burn, or infection, which may arouse the interest of the researchers in the relevant field. In my view, this review can be published for the publication in International Journal of Mechanical Sciences after several revisions, and the suggestions are mentioned below:

  1. The concise summarization of this review should be supplemented in the Introduction section.
  2. The content in Table 1 should be further aligned.
  3. The expression “Most The microbiome can be divided into oral microbiome” in Line 279 is incorrect.
  4. The content of “Institutional Review Board Statement” in Line 457 is repeated from Line 456, which should be deleted during revision.
  5. Some minor errors: 1) the first letter of the word “non-lethal” in Line 17 should be capitalized; 2) the phrase “40% -65%” should be written as “40-65%”; 3) the phrase “Il-10” should be corrected to “IL-10”.
  6. The format of the Reference section should be further checked and improved.

Author Response

Reviewer 3

Comments and Suggestions for Authors

Review Comments

This review provides a comprehensive description of the complicated nature of radiation injury followed by trauma such as hemorrhage, wound, burn, or infection, which may arouse the interest of the researchers in the relevant field. In my view, this review can be published for the publication in International Journal of Mechanical Sciences after several revisions, and the suggestions are mentioned below:

  1. The concise summarization of this review should be supplemented in the Introduction section.
  2. The content in Table 1 should be further aligned.
  3. The expression “Most The microbiome can be divided into oral microbiome” in Line 279 is incorrect.
  4. The content of “Institutional Review Board Statement” in Line 457 is repeated from Line 456, which should be deleted during revision.
  5. Some minor errors: 1) the first letter of the word “non-lethal” in Line 17 should be capitalized; 2) the phrase “40% -65%” should be written as “40-65%”; 3) the phrase “Il-10” should be corrected to “IL-10”.
  6. The format of the Reference section should be further checked and improved.

Authors’ responses

We greatly appreciated reviewer 3’s comments and suggestions that have incorporated into the revised manuscript. The modified areas are highlighted for you to review.

  1. The concise summarization of this review should be supplemented in the Introduction section.

Response: Thanks. The concise summarization of this review was inserted in the Introduction section. Please see lines 47-49.

  1. The content in Table 1 should be further aligned.

Response: Thanks. We further aligned them.

  1. The expression “Most The microbiome can be divided into oral microbiome” in Line 279 is incorrect.

Response: Thanks. We corrected it. Please see line 312.

  1. The content of “Institutional Review Board Statement” in Line 457 is repeated from Line 456, which should be deleted during revision.

Response: Thanks. We deleted it. Please see line 493.

  1. Some minor errors: 1) the first letter of the word “non-lethal” in Line 17 should be capitalized; 2) the phrase “40% -65%” should be written as “40-65%”; 3) the phrase “Il-10” should be corrected to “IL-10”.

Response: Thanks. We made the corrections. Please see lines 34 and 190.

  1. The format of the Reference section should be further checked and improved.

Response: Thanks. We updated the Reference section.